# Modulating Individual Alpha Frequency through Short-Term Neurofeedback for Cognitive Enhancement in Healthy Young Adults

**DOI:** 10.3390/brainsci13060926

**Published:** 2023-06-08

**Authors:** Ben-Zheng Li, Wenya Nan, Sio Hang Pun, Mang I. Vai, Agostinho Rosa, Feng Wan

**Affiliations:** 1State Key Laboratory of Analog and Mixed Signal Very-Large-Scale Integration (VLSI), University of Macau, Macau 999078, China; benzheng.li@cuanschutz.edu (B.-Z.L.);; 2Department of Electrical and Computer Engineering, Faculty of Science and Engineering, University of Macau, Macau 999078, China; 3Department of Physiology and Biophysics, University of Colorado Anschutz Medical Campus, Aurora, CO 80045, USA; 4Department of Electrical Engineering, University of Colorado Denver, Denver, CO 80204, USA; 5Department of Psychology, Shanghai Normal University, Shanghai 200234, China; 6LaSEEB-System and Robotics Institute, LarSys, 1049-001 Lisbon, Portugal; 7Centre for Cognitive and Brain Sciences, Institute of Collaborative Innovation, University of Macau, Macau 999078, China

**Keywords:** neurofeedback training, individual alpha frequency, cognitive performance, alpha oscillation, peak performance

## Abstract

Human alpha oscillation (7–13 Hz) has been extensively studied over the years for its connection with cognition. The individual alpha frequency (IAF), defined as the frequency that provides the highest power in the alpha band, shows a positive correlation with cognitive processes. The modulation of alpha activities has been accomplished through various approaches aimed at improving cognitive performance. However, very few studies focused on the direct modulation of IAF by shifting the peak frequency, and the understanding of IAF modulation remains highly limited. In this study, IAFs of healthy young adults were up-regulated through short-term neurofeedback training using haptic feedback. The results suggest that IAFs have good trainability and are up-regulated, also that IAFs are correlated with the enhanced cognitive performance in mental rotation and n-back tests compared to sham-neurofeedback control. This study demonstrates the feasibility of self-regulating IAF for cognition enhancement and provides potential therapeutic benefits for cognitive-impaired patients.

## 1. Introduction

The connection between human electroencephalogram (EEG) alpha oscillation (~10 Hz) and cognitive processes has been a topic of interest for several decades [1,2,3,4,5,6]. Alpha oscillations have been identified as a reflection of cortical processes involved in top-down inhibitory control and timing [7,8,9]. The large inter-individual variability in alpha oscillation has necessitated the adjustment of the alpha band for individual subjects based on their dominant alpha frequency [1]. This anchor frequency is known as the individual alpha frequency (IAF) or individual alpha peak frequency. The IAF is typically defined as the frequency that provides the highest power in the alpha band ranging from 7 to 13 Hz.

The IAF has been extensively investigated as an EEG biomarker, revealing high heritability [10,11], strong inverse correlation with age after adulthood [12,13], and a shift to lower frequencies in cognitive-related neuropathological alterations such as mild cognitive impairment, Alzheimer’s disease, and various types of dementia [14,15,16,17,18]. Positive correlations between IAF and cognitive processes, including working memory [19,20,21] and attention [8,22,23], motor control [24], and general intelligence [25], suggest that up-regulating IAF to higher frequencies could enhance cognitive performance. Such an approach may alleviate cognitive symptoms in patients and enable peak performance in healthy individuals.

The modulation of individual alpha activities has been explored through various techniques, including neurofeedback [26,27,28,29,30,31,32,33,34,35], repetitive transcranial magnetic stimulation (rTMS) [36,37,38], transcranial alternative current stimulation (tACS) [39], and rhythmic visual stimulus [40]. Among these modulation approaches, neurofeedback, a kind of non-invasive biofeedback that realizes self-regulation of brain functions by measuring neural activities and providing feedback signals, has been the most widely employed, since it can achieve endogenous neuromodulation with few side effects [41].

Numerous studies have reported improved cognitive performance by upregulating individual upper alpha power [26,27,29,34]. However, direct modulation of the IAF by shifting the peak frequency on the EEG spectrum remains underexplored compared to power-related modulation. Angelakis et al. conducted a pilot study using neurofeedback to modulate IAF, demonstrating the feasibility of self-regulating IAF for cognition enhancement in three elderly participants [27]. Despite this, given the typically lower IAF and the limited sample sizes of previous studies in elderly populations, whether neurofeedback can modulate IAF in healthy young adults and whether this leads to enhanced cognitive performance remains unclear.

The present study aimed to investigate the modulatory effect of a neurofeedback system on IAFs in healthy young adults. Specifically, the neurofeedback system extracted real-time IAFs from EEG spectra and provided haptic feedback for participants to self-regulate IAF. The results revealed that IAFs are highly trainable via the two-day short-term neurofeedback training compared to the sham-neurofeedback controls. Moreover, the up-regulation of IAFs was found to be positively correlated with enhanced cognitive performance in mental rotation and n-back tests, suggesting the potential of IAF modulation for improving cognitive functioning.

## 2. Methods

### 2.1. Participants

This study recruited 33 college-aged subjects, 32 of whom completed all sessions. All the participants had normal or corrected vision, were medication-free, and reported no history of neurological or psychiatric disease. They provided written informed consent and received financial compensation for their participation. The research protocol was carried out in accordance with the Declaration of Helsinki and approved by the local research ethics committee.

Participants were randomly assigned to either the experimental group (NF group: 16 subjects, 13 males and 3 females, mean age: 22.12 ± 3.31 years, range: 18–33) or the sham control group (Sham group: 16 subjects, 10 males and 6 females, age: 21.19 ± 2.56 years, range: 18–27). Both groups received identical experiment procedures and instructions, except for the neurofeedback training session. In this session, the experimental group received feedback based on their EEG signals, while the control group received sham feedback from an experimental group participant whose IAF was successfully up-regulated during training.

### 2.2. Experimental Design

The present study employed a short-term, two-day experimental design, with participants completing five short neurofeedback training sessions per day. Baseline EEG measures were recorded prior to and following the training sessions. Behavioral modification was evaluated through cognitive tests conducted on the day following the first baseline measure and the day following the final baseline session. The experimental timeline is illustrated in Figure 1.

During the EEG baseline-recording session, participants were instructed to keep their eyes closed and avoid any eyeball movements. This session consisted of four epochs, each comprising 30 s recording periods separated by 10 s rest periods. During the neurofeedback training session, participants were provided with five training blocks, each containing three one-minute training epochs, separated by 10 s of rest. Participants were instructed to sit in a comfortable position, close their eyes, and place their two index fingers on a haptic feedback device, a pair of vibration motors controlled by the recording computer. Participants were informed that the vibration amplitude changed based on their brain activities and were instructed to find the optimal mental strategy that maximized the vibrating amplitude, making the vibration as large and long-lasting as possible. The vibrating frequency was fixed at 100 Hz, and the vibrating amplitude was moderate and had been tested by each participant before the training sessions to ensure that different vibrating levels were distinguishable and perceptible.

Following completion of the post-training cognitive tests, participants were asked to complete a set of questionnaires to evaluate their perceived levels of fatigue, any potential adverse side effects, and the effectiveness of their mental strategies during the neurofeedback training sessions. Fatigue levels were assessed using the Chalder Fatigue Scale [42], with participants asked to report their current level of fatigue in comparison to their pre-experiment baseline state.

### 2.3. Data Acquisition

During the experiment, participants sat comfortably in a quiet, dimly lit room, and their EEG signals were recorded using 16 Ag/AgCl electrodes placed on an EEG cap according to the international 10–20 system. The electrode locations were O1, Oz, O2, P3, Pz, P4, C3, Cz, C4, T3, T4, F7, F3, Fz, F4, and F8. All recording electrodes were referenced to the left mastoid and grounded to the FP2. The impedances of each electrode were kept below 10 kΩ. The EEG signals were amplified by a bio-signal amplifier (g.USBamp, Guger Technologies, Graz, Austria) with a sampling rate of 256 Hz through a band-pass filter from 2 to 30 Hz, as well as a notch filter set to the unitality frequency to reduce power line interference.

### 2.4. Training Protocol

In the NF training session, the self-regulated parameter was the mean IAF, computed from the occipitoparietal area (channels: P3, Pz, P4, O1, Oz, O2) in the eyes-closed condition due to the previous observation that alpha activities have higher power at occipitoparietal region especially under the eyes-closed condition [1,26].

The IAF was extracted from each recording channel using a sliding fast Fourier transform algorithm. A Hanning window with a length of 2 s, a 95% overlap, and 6 s of zero padding were used. The resulting IAF values were smoothed using a Savitzky-Golay filter. The IAF was defined as the local maximum within the smoothed alpha-band spectrum (7–13 Hz), with a frequency resolution of 0.125 Hz, and was updated every 100 ms during the training for the neurofeedback. If the alpha peak was not distinctive due to low alpha power or motion artifacts, the IAF feedback amplitude would not update.

The haptic feedback amplitude was computed based on the mean IAFs from the occipitoparietal region, and manual thresholds were assigned according to the IAFs in previous EEG baseline or training sessions. The vibration amplitude was calculated by accumulating the net increase of real-time IAF that exceeded the threshold for a duration of 500 ms. Initially, the threshold frequency was set at 0.1 Hz lower than the IAF obtained in the eyes-closed EEG baseline and was adjusted after each training session.

To assess the training progress, the ratio of sliding-window time steps with real-time IAF exceeding the threshold frequency to the total number of time steps was calculated after each training block. If the real-time IAF was higher than the threshold frequency for more than 70% of the time in the last training block, the threshold frequency would be increased by 0.1 to 0.3 Hz. Conversely, if the real-time IAF was lower than the threshold frequency for more than 30% of the time, the threshold frequency would be decreased by 0.1 to 0.3 Hz.

### 2.5. Cognitive Tests

In the cognitive assessment sessions, the participants began with one block of the 1-back test, followed by two blocks of the 3-back test, and concluded with two blocks of the mental rotation test. Prior to taking the tests, participants were given detailed instructions on how to respond to the stimuli by pressing two buttons on a keyboard with their right index finger and right middle finger, and were provided with sufficient practice opportunities to ensure familiarity with the task requirements.

The mental rotation test implemented in this study was developed based on two previously established versions of mental rotation tests, namely IST70 [43] and A3DW [44]. Each trial of the MR test consisted of a 3-s central fixation cross followed by 8 s of stimulus presentation, as shown in Figure 2. Participants were instructed to indicate whether the displayed pair of cubic dice were congruently matched (Figure 2a) or mismatched (Figure 2b) by pressing designated buttons as quickly and accurately as possible. The test was administered in two blocks per day, each comprising 15 trials arranged in a pseudo-random sequence with a 50% chance of presenting matched pairs. Performance was evaluated by calculating the accuracy, reaction time, and reaction control, defined as the standard deviation of the single-trial reaction time.

The N-back tests were conducted with visuospatial stimuli [45,46], which consisted of light blue squares appearing at one of eight slotted positions around a central fixation cross on a black background (Figure 3). Each trial began with a fixation cross displayed at the center of the screen for two seconds, followed by the stimulus presented for 250 ms. Subjects were instructed to respond by pressing a button with their right index finger only when a matching target had been presented one (1-back) or three (3-back) trials before. The 3-back test included two blocks with 30 trials each, and the 1-back test included one block with 28 trials. Stimuli were presented in pseudo-random sequences with a 33% probability of presenting a matching target. Cognitive performance was evaluated by measuring reaction time and reaction control, and accuracy was also calculated for the 3-back test by summing the hits (number of targets minus number of missed targets) and correct rejections (number of distractors minus number of false alarms) and dividing by the total number of trials.

### 2.6. Data Processing and Statistical Analysis

In the offline data analysis, the IAFs for each session were computed by determining the average IAF across all recording epochs. The EEG recordings over the epoch were first segmented by a 2-s 50% overlapping sliding window, and then detrended by Savitzky–Golay filter and zero-padded to reach a frequency resolution of 0.01 Hz. The IAFs were calculated as the center of gravity frequency in the alpha band (7–13 Hz). The amplitude of the individual alpha band was estimated by computing the power ratio between individual alpha bands (IAF ± 1 Hz) and the overall spectrum (3–30 Hz). The IAF heatmaps were generated by using the Talairach coordinate [47] and the Mitchell-Netravali filter for interpolation.

The Shapiro-Wilk test was used to confirm the normality of the data, and non-parametric methods were used when the normality was not satisfied, such as behavioral-performance metrics. The initial IAFs during the first baseline session between the two groups were compared using the independent-samples *t*-test. The change of IAFs over all recording sessions was examined using mixed analysis of variance (ANOVA), with the between-subject factor as group (NF vs. Sham) and within-subject factor as time (total of 14 baselines and training blocks). A pairwise multiple comparison with Holm-Bonferroni correction was performed to determine the changing tendency of IAFs. The initial behavioral performances between the two groups were compared using a Mann-Whitney U test, and the consequent changes were analyzed using the Friedman test, which is a non-parametric alternative to the one-way ANOVA for repeated measures. For each subject, the behavioral score was computed as the count of improved behavioral metrics before and after training sessions. Spearman’s correlations were performed between the behavioral scores and the cumulative changes of baseline IAFs for each day. The data analysis was performed using custom python scripts with Scipy [48] and Pingouin [49] libraries.

## 3. Results

### 3.1. Up-Regulated IAFs

Eyes-closed baseline IAFs and alpha ratios prior to the training session are displayed in Figure 4. Subjects from both groups exhibited similar IAFs (Figure 4a; T(30) = 0.839, *p* = 0.408) and individual alpha ratios (Figure 4b; T(30) = −0.357, *p* = 0.723), with a high degree of overlap in the averaged spectra (Figure 4c).

Through neurofeedback training, IAFs changed over sessions for both groups, as depicted in Figure 5a. Consistent with the training objective, a mixed ANOVA revealed a significant main effect of time (F(1,30) = 6.1812, *p* < 0.0001) and significant time × group interaction (F(13,390) = 1.9515, *p* = 0.0236). Pairwise comparisons demonstrated significantly increased IAFs in the NF group during the training sessions (pre baseline vs. NF block 10: T(15) = −5.854, *p* = 0.0027) and post-training baselines (pre baseline vs. post baseline: T(15) = −4.303, *p* = 0.047), while no significant changes were observed in the Sham group. The upregulation of IAFs was further illustrated by the averaged EEG baseline spectrum at Oz, where neurofeedback training shifted the individual alpha peak (Figure 5b) and sham training did not clearly cause changes in the spectrum (Figure 5c).

The self-regulation of occipitoparietal IAFs was accompanied by effects on other brain regions, as demonstrated in the heatmaps in Figure 6. Compared to the initial baseline, neurofeedback training increased IAFs by approximately 0.182 Hz evenly across all brain regions during the final training session. However, during the post-training baseline session, relatively larger changes in IAFs were observed near the central sulcus, with approximately a 0.125 Hz increase at C3, Cz, C4, and around a 0.0859 Hz increase in the remaining brain regions. The central IAFs exhibited a higher divergence between two groups over sessions (Figure 7a) with more significant time × group interaction (mixed ANOVA: F(13,390) = 2.1939, *p* = 0.0093) compared to the change in overall IAFs shown in Figure 5a.

The power ratio of the individual alpha band (Figure 7b) exhibited a significant change with time in a mixed ANOVA (F(13,390) = 7.975, *p* < 0.0001), but there was no significance in time × group interaction (F(13,390) = 1.316, *p* = 0.2002). Pairwise comparisons revealed a significantly-attenuated alpha ratio between baselines and training sessions in the NF group only (pre baseline vs. NF block 2: T(15) = 4.619, *p* = 0.0274), while no significant differences were observed between baseline sessions in either group.

### 3.2. Enhanced Behavioral Performance

Subjects in both groups had similar initial performance in the mental rotation test for accuracy (Figure 8a; U = 117.5, *p* = 0.705) and reaction time (Figure 8b; U = 155.0, *p* = 0.318). After the training sessions, improved mental rotation accuracy was observed in the NF group (Figure 8c left; Q = 4.571, *p* = 0.0324) but not in the Sham group (Figure 8c right; Q = 0.067, *p* = 0.796). Both groups exhibited a significant reduction in reaction time (Figure 8d; NF and Sham: Q = 6.25, *p* = 0.0124).

For the n-back test, no significant differences were found in the initial 1-back reaction time (Figure 9a: reaction time: U = 118, *p* = 0.72), 3-back accuracy (Figure 9b: U = 108, *p* = 0.462), or 3-back reaction time (Figure 9c: U = 179, *p* = 0.054). Neither group exhibited significant improvements in 1-back reaction time after training (Figure 9d: Q = 0, *p* = 1.0). However, both groups showed improved 3-back accuracy (Figure 9e: NF: Q = 6.25, *p* = 0.0124; Sham: Q = 7.143, *p* = 0.00752) and shorter 3-back reaction times (Figure 9f: NF: Q = 4, *p* = 0.0455; Sham: Q = 6.25, *p* = 0.0124).

Considering all task metrics together, the overall behavioral score, which is calculated as the total count of improved task metrics for individual subjects, was found to be positively correlated with subjects’ cumulative change in IAFs in the NF group only (Figure 10a: NF: r = 0.6355, *p* = 0.00814; Figure 10b: Sham: r = −0.0227, *p* = 0.9333). No direct correlation was found between the percentage change of behavioral metrics and the percentage change of IAFs.

### 3.3. Fatigue and Adverse Side Effects

Subjective fatigue scores indicated that most subjects did not become tired from the experiment. The majority of subjects reported their fatigue levels as “no more than usual”. Three subjects in the NF group reported experiencing sleepiness much more than usual. There was no statistical difference between the fatigue levels after training in the two groups (Mann-Whitney U test: U = −1.537, *p* = 0.142).

No adverse side-effects were reported by 22 out of 32 participants. The most frequently reported adverse side-effects after the experiment were itchy skin from six subjects, including five from the NF group and one from the Sham group. One subject from each group reported experiencing a slight headache during short periods. No significant differences were found between the two groups in the rating of side effects (Mann-Whitney U test: U = −0.879, Exact *p* = 0.467).

### 3.4. Mental Strategies

Recorded effective strategies were categorized into three types based on emotional valences, i.e., positive (pleasant), neutral, and negative (unpleasant). Positive types commonly included concepts such as friends, family, entertainment, love, etc. Neutral types comprised math calculations, recitations, counting numbers, etc. The negative type consisted of sorrow, anger, quarrels, fear, etc. In the NFT group, 13 subjects listed positive strategies, 10 subjects listed neutral strategies, and 1 subject listed negative strategies. In the Sham group, 8 subjects listed positive strategies, 8 subjects listed neutral strategies, and 5 subjects listed negative strategies. In contrast, although most subjects were inclined to use pleasant mental strategies during training, more pleasant strategies and fewer unpleasant strategies were provided by subjects who received real NF than those who received Sham feedback, as shown by the percentages listed in Figure 11. This implies higher occurrence of positive mental strategies in neurofeedback training for up-regulating IAF.

## 4. Discussion

The presented results demonstrate the feasibility of IAF upregulation through neurofeedback, with good trainability that shifted resting-state baseline IAFs by 0.2 Hz through a total of 30 min of training sessions. This suggests greater flexibility of IAFs in young healthy adults compared to elderly people, for whom previous work reports a shift of 0.5–0.7 Hz through weeks of training, totaling around 15 h [27]. The resilience of the trained IAF is also apparent between day-1 post-training baselines and day-2 pre-training baselines, which implies higher flexibility and suggests that IAF neurofeedback studies with long-term training sessions and follow-up measurements are needed to further assess trainability and estimate the saturation and ceiling levels.

Subjects in the Sham group exhibited significant improvement in cognitive test performance, although they did not upregulate IAFs with irrelevant feedback. This improvement could be a result of training effects on the cognitive tests, or placebo effects, as neurofeedback may offer a potent psychological intervention and produce a super placebo effect [50], which could give participants more confidence in making decisions in cognitive tests. Therefore, as is frequently advocated in the field [51], sham neurofeedback controls are necessary to separate the real behavioral modification outcomes from the trained protocol. Subjects in the NF group not only demonstrated similar improvement in the metrics that improved in the Sham group, such as faster reaction speed, but also showed significantly enhanced accuracy in mental rotation tests and a positive correlation between IAF increments and overall behavioral scores. These differences between the two groups could be regarded as relative cognition enhancement through IAF neurofeedback training above the placebo and training effects.

It is intuitive to think that direct IAF modulation could be mathematically equivalent to other alpha modulation protocols, such as up-regulating upper alpha power [26,28,29,31,34] or down-regulating theta(4–7 Hz)/beta(13–30 Hz) power ratio [52,53,54]. However, the consequent effects of these protocols on IAFs showed discrepancies, where some studies reported significant increases in IAFs after training [34,53], others reported no significant changes in IAFs despite significant increases in upper alpha power [29,31]. In contrast, the direct IAF modulation in our study showed increased IAFs with no changes in alpha power and upper alpha power after training, and even significantly reduced power during the training. This strongly suggests that different underlying neural mechanisms may exist between various modulation protocols.

The modulation efficacy in the presented IAF neurofeedback may also be facilitated by the haptic feedback scheme, which is relatively rare in traditional neurofeedback studies and occasionally employed in research related to motor imagery and stroke rehabilitation [55,56]. Subjects receiving haptic feedback could maintain better attention and concentration during training, possibly because the boredom resulting from constant eye contact in visual feedback and distraction or anxiety in auditory feedback can be effectively avoided [57]. The type of feedback might also be connected to the underlying neural activities; for example, in our results, heat maps indicated the most significant changes in trained IAF were from central electrodes (Cz, C3, C4), which are directly above the somatosensory cortex and may be more sensitive to haptic information.

Participants in the NF group exhibited a significant increase in IAF and cognitive performances relative to the Sham group through short-term neurofeedback training. This investigation is the first study demonstrating the trainability of IAF through neurofeedback and verifying the feasibility of IAF modulation for cognitive enhancement in healthy young adults. This thereby aligns with the conclusions of the previous pilot study of IAF neurofeedback in three elderly individuals [27]. However, it is important to consider certain limitations that may constrain the generalizability and extrapolation of these results. The relatively short training time and the limited number of training sessions may not adequately predict the long-term consequences of IAF neurofeedback in healthy young adults, highlighting the need for more extended investigations. Additionally, the Sham group exhibited learning effects in cognitive tests, indicating that future studies should incorporate a wider selection of cognitive tests to further examine which cognitive categories are specifically sensitive to IAF neurofeedback. Successfully overcoming these limitations could contribute to the development of practical applications using IAF neurofeedback for cognitive enhancement and the possibility of clinical applications that could help alleviate symptoms of cognitive impairments.

## 5. Conclusions

This study explored the feasibility and effectiveness of upregulating IAF through neurofeedback training in young healthy adults to enhance cognitive performance. The results demonstrated that IAFs were significantly increased through two days of short-term training. Neurofeedback training also improved cognitive performance in both the neurofeedback and sham groups; however, the improvement in the neurofeedback group was more pronounced and correlated with the upregulation of IAFs. These findings suggest that IAF modulation could be an effective neurofeedback training protocol for cognitive enhancement.

## Figures and Tables

**Figure 1 brainsci-13-00926-f001:**
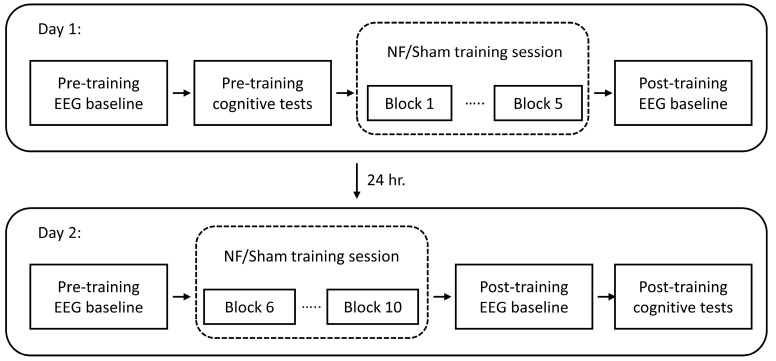
Schematic of the experimental workflow.

**Figure 2 brainsci-13-00926-f002:**
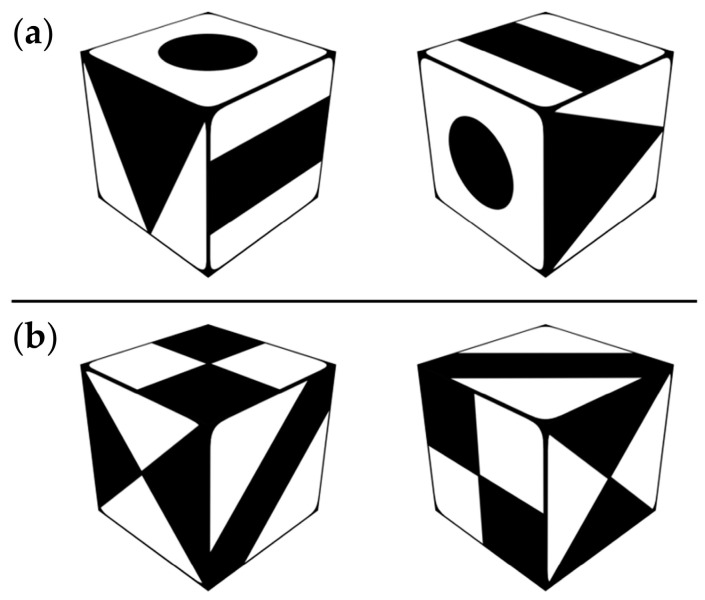
Examples of a matching pair (**a**) and a mismatching pair (**b**) of dice in the mental rotation test.

**Figure 3 brainsci-13-00926-f003:**
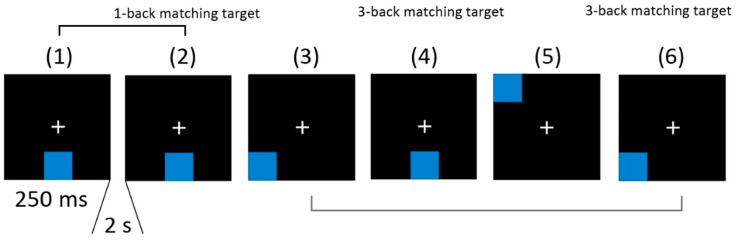
The sequence of stimuli of a visuospatial n-back tests (1-back and 3-back).

**Figure 4 brainsci-13-00926-f004:**
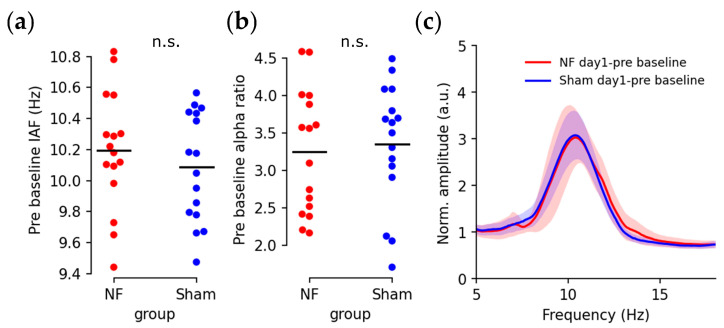
Initial individual alpha activities measured in the day-1 pre-training baseline: (**a**) Initial IAF (n.s.: *p* > 0.05; red dot: NF group; blue dot: Sham group; black bar: average); (**b**) Initial individual alpha power ratios; (**c**) Mean EEG (Oz) spectrum of initial baselines with the scale normalized by the overall band power (red line: NF group; blue line: Sham group; shading: standard error).

**Figure 5 brainsci-13-00926-f005:**
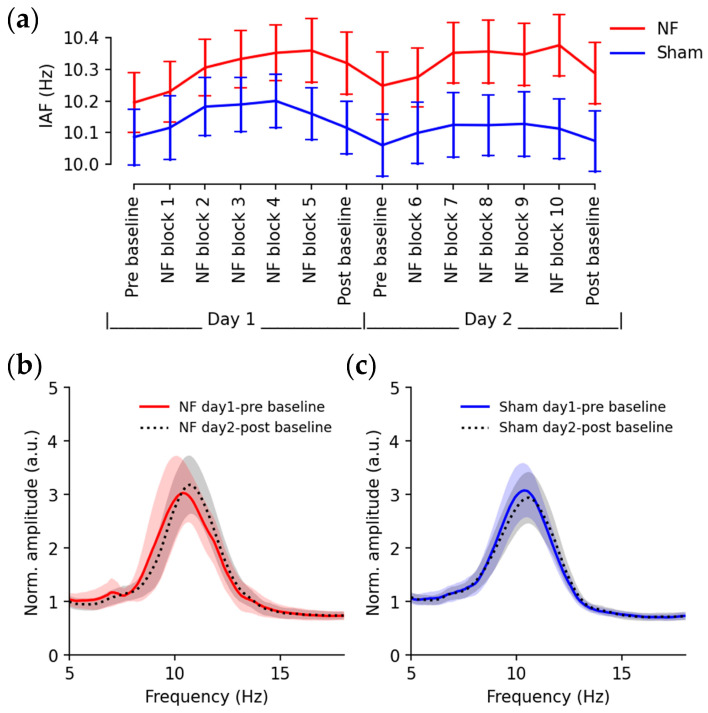
Up-regulation of IAFs through neurofeedback training (**a**) IAFs during baseline and training sessions (error bars: standard errors); (**b**) averaged spectrum of the day-1 pre-training baseline (red line) and day-2 post-training baseline (black dash line) from subjects in the NF group; (**c**) averaged spectrum of the day-1 pre-training baseline (blue line) and day-2 post-training baseline (black dashed line) from subjects in the Sham group (shading: standard error).

**Figure 6 brainsci-13-00926-f006:**
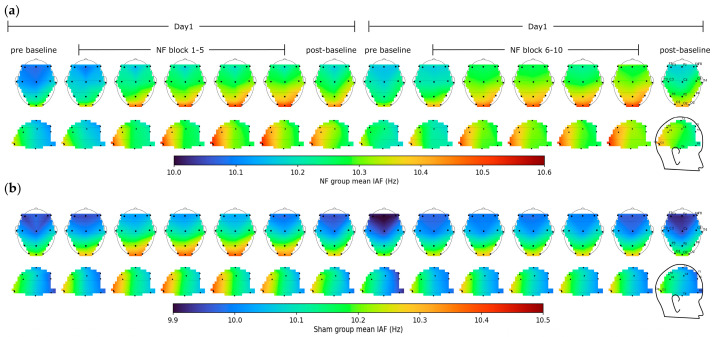
EEG heat maps of mean IAFs across baseline and training sessions in the NF group (**a**) and Sham group (**b**).

**Figure 7 brainsci-13-00926-f007:**
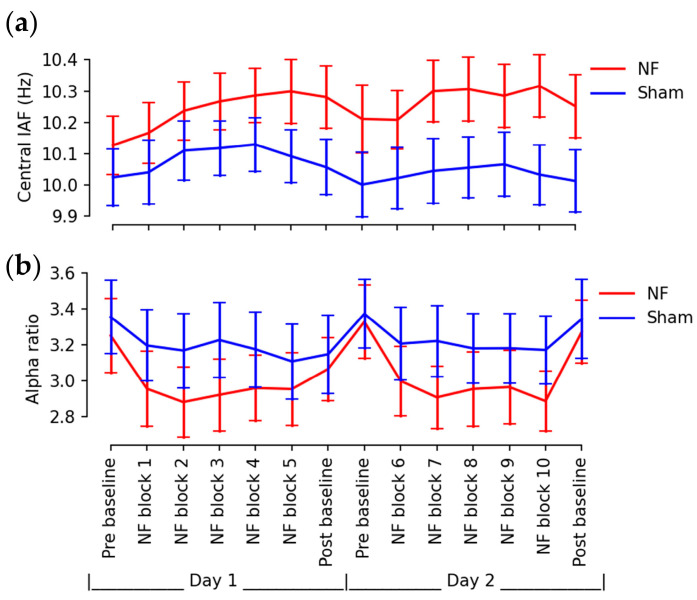
(**a**) Central IAF and (**b**) overall alpha ratio during baseline and training sessions (error bars: standard errors).

**Figure 8 brainsci-13-00926-f008:**
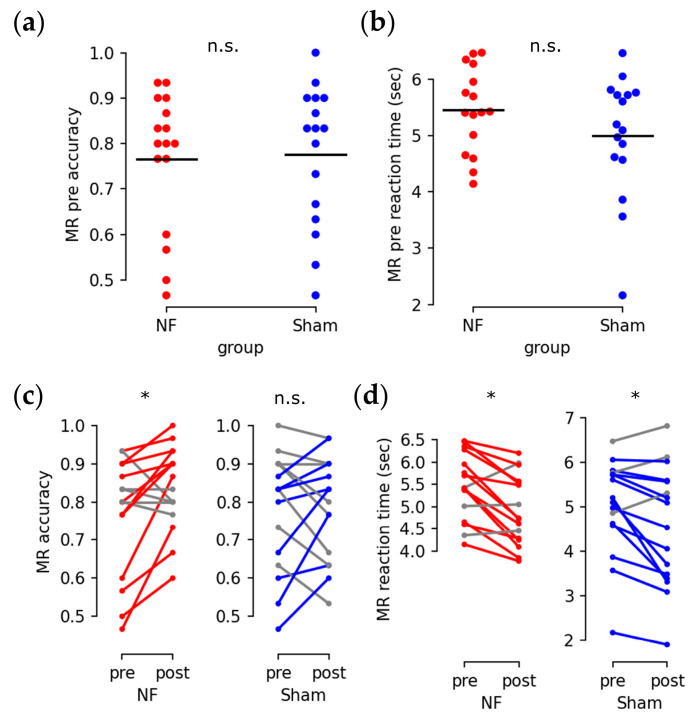
Behavioral performances of the mental rotation test. (**a**,**b**) initial accuracy (**a**) and reaction time (**b**) of the NF group (red dot) and the Sham group (blue dot) (n.s.: *p* > 0.05; black bar: average). (**c**,**d**) before and after training change in accuracy (**c**) and reaction time (**d**) (*: *p* < 0.05; red segments: improved pair for NF group subjects; blue segments: improved pair for Sham group subjects; grey segments: unimproved pairs).

**Figure 9 brainsci-13-00926-f009:**
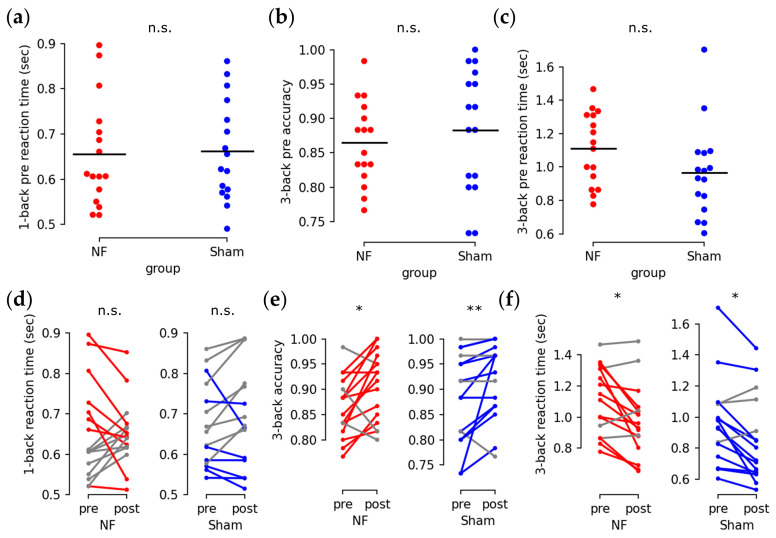
Behavioral performances of n-back test. (**a**–**c**) initial 1-back test reaction time (**a**), 3-back test accuracy (**b**) and 3-back test reaction time (**c**) between the NF group (red dot) and the Sham group (blue dot) (n.s.: *p* > 0.05; black bar: average). (**d**–**f**) before and after training change in 1-back test reaction time (**d**), 3-back test accuracy (**e**) and 3-back test reaction time (**f**) (*: *p* < 0.05; **: *p* < 0.01; red segments: improved pair for NF group subjects; blue segments: improved pair for Sham group subjects; grey segments: unimproved pairs).

**Figure 10 brainsci-13-00926-f010:**
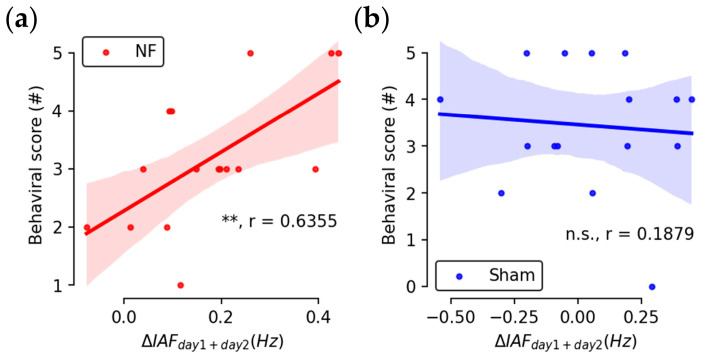
Scatter plot and linear regression between counts of improved behavioral metrics and cumulative changes of baseline IAFs for subjects in the NF group (**a**) and the Sham group (**b**) (n.s.: *p* > 0.05; **: *p* < 0.01; r: correlation coefficient; shading: 95% confidence interval).

**Figure 11 brainsci-13-00926-f011:**
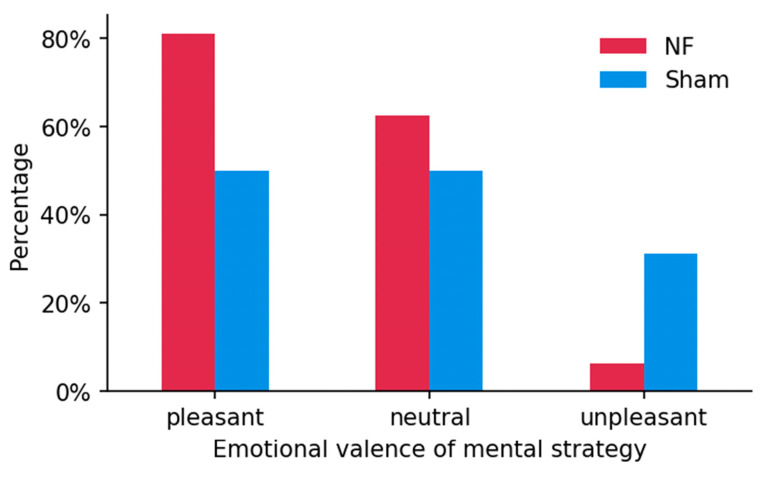
Percentage of reported mental strategies with different emotional valence between two groups.

## Data Availability

The experimental data and analysis scripts are available upon reasonable request.

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
