# Peer review of "Modulating Individual Alpha Frequency through Short-Term Neurofeedback for Cognitive Enhancement in Healthy Young Adults"

_brainsci, 2023, doi:10.3390/brainsci13060926_

Round 1

Reviewer 1 Report

The manuscript is written logically and clearly. Meanwhile, I have several questions for the authors:

It’s original and interesting approach of using haptic feedback. Could authors validate of their choice.

 Why did they prefer this feedback modality in comparison with the sound?

 What the hypotheses was to put  the initial threshold frequency  at 0.1 Hz lower than the IAF in the eyes-closed EEG baseline?

Could authors provide explanation of what was the paradigm to obtain the thresholds?

How did authors calculate the frequency of appearance the iAF above the threshold?

I suggest to use the moderate editing of English language.

Reviewer 2 Report

In this study, authors reported the differences modulating individual alpha frequency through short-term neurofeedback for cognitive enhancement in healthy adults. The overall contents of this manuscript is well organized to give a clear overview of this work. I have suggested some comments about this work are as the following:

Comments to the Authors:

1. Authors should write clearly abstract including background, method, results with numerical values of delta (1-3Hz), theta (4-7Hz), alpha (8-12Hz), beta (13-30Hz) and gamma (30-50HZ) power.

2. Authors should revise the introduction section in three paragraph, first paragraph for EEG-based studies that modulating individual alpha frequency through short-term neurofeedback for cognitive enhancement, second paragraph for past research based individual alpha frequency through short-term neurofeedback using EEG/fMRI/MEG, third paragraph for gap between past and present results and in last paragraph objective of this study and hypothesis.

3. In method section author should write the statistical methods about the analysis clearly. Authors can refers the following papers: Chikara RK, Chang EC, Lu Y-C, Lin D-S, Lin C-T and Ko L-W (2018) Monetary Reward and Punishment to Response Inhibition Modulate Activation and Synchronization Within the Inhibitory Brain Network. Front. Hum. Neurosci. 12:27. doi: 10.3389/fnhum.2018.00027

4. My suggestion is that the authors should write discussion section clearly in more details like how and why this study is important than previous studies based on individual alpha frequency through short-term neurofeedback for cognitive enhancement estimation using EEG.

5. The authors should write some limitations of this study and future clinical application in more details.

Minor editing of English language required.
